# Influence of Monomer Size on CO_2_ Adsorption and Mechanical Properties in Microporous Cyanate Ester Resins

**DOI:** 10.3390/polym17020148

**Published:** 2025-01-09

**Authors:** Yukun Bai, Gota Kikugawa, Naoki Kishimoto

**Affiliations:** 1Department of Chemistry, Graduate School of Science, Tohoku University, Aramaki, Aoba-ku, Sendai 980-8578, Japan; 2Institute of Fluid Science, Tohoku University, Katahira, Aoba-ku, Sendai 980-8577, Japan

**Keywords:** cyanate resin, molecular dynamics simulation, thermosetting materials, CO_2_ adsorption, crosslinking model

## Abstract

Molecular simulations offer valuable insights into thermosetting polymers’ microstructures and interactions with small molecules, aiding in the development of advanced materials. In this study, we design two cyanate resin models featuring monomers of different sizes and employ a previously developed method to generate crosslinked structures. We then analyze their crosslinking processes and physicochemical properties. Using quantum chemistry calculations and a GCMC/MD approach, we investigate CO_2_ adsorption. Our results show that monomer size does not significantly affect the crosslinking process and provides a degree of polymerization as 83.8 ± 0.3% vs. 76.7 ± 1.4%, but it does influence key properties, such as the glass transition temperature (520 K vs. 420 K) and Young’s modulus (2.32 GPa vs. 1.77 GPa). Moreover, CO_2_ adsorption differs between the two models: the introduction of propyl ether moieties lowers by around 70% CO_2_ uptake, indicating that specific adsorption sites impact gas adsorption. This study demonstrates a promising strategy for designing and optimizing thermosetting polymers with controllable gas separation and storage capabilities.

## 1. Introduction

Thermosetting polymer materials—particularly cyanate ester resins—exhibit outstanding physical and chemical properties such as high glass transition temperatures and high Young’s moduli [1,2,3]. These characteristics have led to their widespread application in various high-tech fields, including the aerospace industry [4], electronic adhesives, and membranes for ionic liquid separations [5], among others. Cyanate monomers can polymerize either in the presence or absence of catalysts. Without a catalyst, the cyanate functional groups (-OCN) undergo a two-step crosslinking process, naturally forming a three-dimensional network with rigid triazine ring structures. This crosslinked architecture introduces microporosity and increases the free volume within the polymer matrix [6].

The polymer framework readily interacts with small organic molecules, including CO_2_, H_2_O, and N_2_ [7,8]. In particular, the dipole–quadrupole interactions arising from nitrogen and oxygen atoms in the triazine rings enhance the affinity of CO_2_ for these sites. As a result, cyanate resins have been considered next-generation materials for carbon capture, thereby contributing to carbon neutrality goals.

Molecular dynamics (MD) simulations offer a cost-effective approach to investigate structure–property relationships at the molecular level—information that is often challenging to obtain experimentally [9,10]. By employing MD simulations, one can readily examine key physical and chemical properties such as Young’s modulus, glass transition temperature, and gas adsorption characteristics. In prior studies, the adsorption positions and uptake values of various gas molecules (e.g., H_2_O and CO_2_) in different cyanate resins have been reported through experimental and simulation methods [11]. In Wang’s group’s research, multitype polycyanurate materials have been developed and characterized experimentally [12,13].

However, previous simulation approaches have several limitations. Many models consider only the spatial arrangement of monomers [14,15], thereby underestimating the realistic reaction probabilities and structural defects by missing the dynamic process during polymer crosslinking. This simplification can lead to models that do not accurately reflect the pore structure of the polymer and which may overestimate gas adsorption capacities [16]. Small model sizes also fail to capture the complexities of the pore network in real polymer materials [11]. Therefore, although previous studies have analyzed gas adsorption, the lack of robust modeling methodologies for constructing realistic three-dimensional cyanate resin structures makes it challenging to understand polymer behavior based on molecular geometry, especially regarding CO_2_ adsorption.

In previous research, we have employed a multistep GRRM/MC/MD modeling approach that our group had previously developed [9,17]. In that approach, the GRRM method [18] is used to perform quantum chemical (QC) calculations that determine reaction energies between functional groups, confirming the formation of triazine rings. Monte Carlo (MC) methods are then employed to incorporate reaction rates into the MD simulations using Arrhenius equations derived from these QC calculations [19]. The details of the crosslinking algorithm are available in the Appendix A. After generating the polymer models, we calculated the glass transition temperature and Young’s modulus. Finally, a combined GCMC/MD simulation framework was used to investigate CO_2_ adsorption in the polymer, and QC calculations were performed to identify CO_2_ adsorption sites.

In this study, we report on two distinct monomers with varying steric hindrance around the cyanate groups, as shown in Figure 1, to understand how monomer structure influences the mechanical and chemical properties of the resulting polymers. The free MD simulation software LAMMPS-3Nov2022 [20] is used for relaxation and for characterizing the physical and chemical properties. We also employ a combination of GCMC/MD methods [21] to predict CO_2_ adsorption capacities. The GRRM method is used for DFT calculations to analyze CO_2_ adsorption sites and calculate the corresponding interaction energies.

## 2. Calculation Methods

### 2.1. Quantum Calculations Method

In this study, we employed the 2020 version of the GRRM program [18], interfaced with Gaussian 16 [22], to optimize monomer structures and calculate CO_2_ adsorption positions and energies. Initially, the monomer geometries were optimized at the B3LYP-D3/6-311+G(d) level of theory using the MIN modules [23], ensuring that we obtained energetically favorable conformations. Partial atomic charges before and after the reaction were determined using the restrained electrostatic potential (RESP) method [24] at the same level of theory. CO_2_ adsorption energies were then calculated following Equation (1).

Additionally, to gain deeper insights into the intermolecular interactions within the cyanate resin, specifically between the newly formed triazine ring and the adjacent benzene rings, we employed the artificial force induced reaction (AFIR) modules [25]. These calculations were performed at the B3LYP-D3/6-311+G(d) level of theory, providing accurate and detailed information on the intermolecular forces in the resin structure.

### 2.2. Modelling Framework

To ensure the accuracy of the simulation results, three independent initial configurations were created for each system, each starting from a randomly generated initial structure. The initial structures were generated using the Winmoster 10 software [26], which employed the cyanate monomer models derived from the QC calculations described previously. The MD simulations utilized the Dreiding force field [27], combined with the Lennard–Jones potential, which is widely used in predicting the thermophysical properties and providing acceptable results [28,29]. All the MD simulations in this research used a free simulation software, LAMMPS-3Nov2022. Each system consisted of 150 cyanate monomers randomly inserted into the simulation box. Two different monomer types were considered: monomer Figure 1a, referred to as S-small, and monomer Figure 1b, referred to as S-large. A timestep of 0.5 fs was used in all MD simulations to maintain structural stability.

Temperature control in the isochoric–isothermal (NVT) ensemble was achieved with a Nosé–Hoover thermostat [30], while both temperature and pressure in the isobaric–isothermal (NPT) ensemble were regulated using the Martyna–Tobias–Klein (MTK) equations of motion [31]. Periodic boundary conditions (PBCs) were applied to all three dimensions. A 12 Å cutoff was set for van der Waals and real-space Coulombic interactions. Long-range Coulombic interactions were calculated using the particle–particle–particle–mesh (PPPM) method [32], with an accuracy parameter of 1.0 × 10^−5^.

Before crosslinking, the systems underwent an annealing process to relax the structures. First, the systems were heated from 300 K to 1000 K over 50 ps and then cooled back down to 300 K over 700 ps under the NVT ensemble. Following this, an NPT ensemble at 300 K and 1 atm was applied for 50 ps to equilibrate the systems, completing one annealing cycle. This annealing cycle was repeated 10 times. Finally, an additional 5 ns of simulation under the NPT ensemble at 300 K was performed to further reduce residual internal stress in the polymer models.

### 2.3. Crosslinking Method

In this study, we employed a multistep cyanate resin modeling approach previously reported by our group [9] and validated it by comparing it with experimental values. The reaction energies for the two monomer crosslinking reactions were considered identical to those in our previous works, and the reaction temperature was set to 473 K at 1 atm. The crosslinking process was allowed to proceed iteratively for up to 300 cycles. To better understand the reaction progression, we defined the degree of polymerization (DOP) as the ratio of the number of reacted nitrogen atoms to the total number of initially reactive nitrogen atoms, as shown in Equation (1), where N is the number of atoms. This metric provided a quantitative measure of the extent of polymerization during the simulation.DOP = N_reacted nitrogen atoms_/N_total nitrogen atoms_ × 100%(1)

### 2.4. Post-Crosslinking Procedure

After the crosslinking procedure, a post-crosslinking relaxation protocol was employed to reduce the residual internal stress and remove energetically unfavorable configurations. First, the models were maintained at 473 K for 50 ps. Subsequently, under an NVT ensemble, the temperature increased from 300 K to 1000 K over 100 ps. Once the system reached 1000 K, it was held at this temperature and 1 atm under NPT conditions for 50 ps. Next, the system was cooled from 1000 K back down to 300 K at a rate of 10^12^ K/s under NVT conditions and then allowed to relax under NPT conditions at 300 K and 1 atm for an additional 100 ps. This heating–cooling–relaxation cycle was repeated five times, from 300 K to 1000 K and back to 300 K. After completing these five cycles, a final NPT ensemble simulation at 300 K and 1 atm for 10 ns was performed to ensure the polymer models were fully relaxed and equilibrated.

### 2.5. Properties Analysis

MD simulations can provide valuable insights into the glass transition temperature (*T*_g_) and Young’s modulus of polymeric systems by analyzing changes in density and volume under varying temperature or pressure conditions. Combined with the grand canonical Monte Carlo (GCMC) method, these simulations can also yield important information regarding gas adsorption behavior. The details of the property evaluation methods are described below.

Glass transition temperature: the glass transition temperature of a cyanate resin, a dynamic property, can be predicted by examining how its density changes differ between the rubbery and glassy states [33,34]. To accurately calculate *T*_g_, the systems were first equilibrated at 200 K for 200 ps and then heated to 900 K at 10^8^ K/s. During this heating process, density changes were recorded. Above *T*_g_, the system enters the rubbery state and its density decreases significantly with increasing temperatures. Below *T*_g_, the system remains in a glassy state, where the density is relatively insensitive to temperature changes. The density–temperature curves were fit using two straight lines corresponding to the glassy and rubbery regions based on the least squares method, following the approach outlined in previous work [35,36] (Figure 2). The calculated root means square errors (RMSEs) for each system are shown in Appendix A.

**Figure 2 polymers-17-00148-f002:**
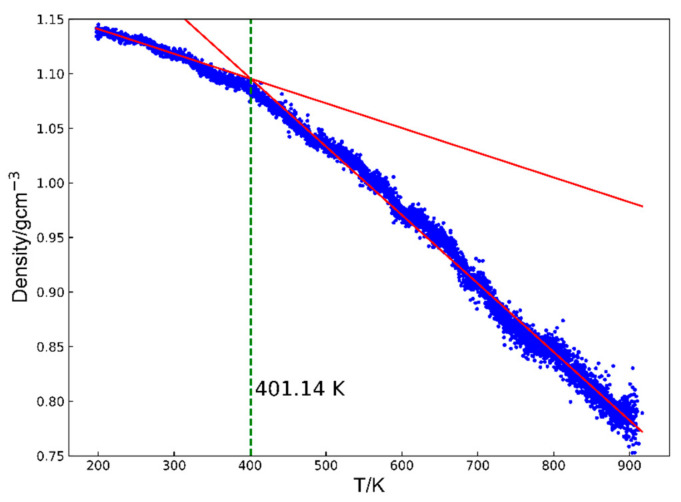
The representative output of the volume–temperature plot for an S-large system consists of two straight lines fitting the curve to represent the two states, glassy and elastic. The cross-point temperature for two lines is determined as the glass transition temperature, as indicated by the green dashed line.

Young’s modulus: the models obtained from the NPT equilibration stage were subjected to uniaxial tensile deformation at a strain rate of 10^7^ s^−1^ along one principal direction. The dimensions in the other two directions were allowed to fluctuate independently under atmospheric pressure conditions (i.e., decoupled boundary conditions). As the model was stretched in one direction, the other directions contracted, and the resulting stress–strain behavior was recorded. From these stress–strain curves, the Young’s modulus was determined by measuring the slope of the linear elastic region up to 5% strain. To obtain a more representative property, the Young’s modulus of each model was averaged over the x, y, and z directions.Free volume: the free volume is calculated by the Multiwfn 3.8 software package [37]. A uniform grid was generated inside the box containing the structures, and each grid point lying outside the van der Waals radii of the probe was classified as void. The free volume was then determined using a grid spacing of 0.25 Å.CO_2_ adsorption: CO_2_ uptake was simulated using a combined GCMC/MD approach at 50 atm and 298 K. The GCMC method allows the insertion and deletion of gas molecules based on energy and chemical potential considerations, thereby identifying energetically favorable sites within the polymer. After each GCMC step, system volume adjustments were made to emulate the real gas uptake process. The CO_2_ molecules were modeled using parameters derived from the quantum chemistry calculations. For each GCMC/MD cycle, 10,000 GCMC trials were performed, followed by 10,000 NPT MD steps. This procedure was repeated for 40 cycles in this study, ensuring that the systems reached equilibrium and were fully saturated with CO_2_.

## 3. Results

### 3.1. Crosslinking Process

During the crosslinking procedure, we controlled both the reaction time and the conversion using our previously developed Python scripts. As reported in earlier studies, these methods ensure the full curing of both systems under consideration. The total number of crosslinking iterations was set to 300 in all cases. The only difference between the two systems lied in the monomer size.

The initial monomer distribution was arranged in an energetically favorable configuration for the S-small system, as illustrated in Figure 3a. After the crosslinking procedure, a fully cured cubic cell with dimensions of approximately 3.9 × 3.9 × 3.9 nm^3^ and containing 150 monomers was obtained, as shown in Figure 3b.

After crosslinking, both systems’ physical and chemical properties were calculated following the methodology described in Section 2.5. The results, averaging over three independent initial configurations for each system, are summarized in Table 1.

To understand the crosslinking process for two systems, the gel point, which is the point of materials fluidity significantly reduced because of the connected structure to form a cluster, was analyzed. In the simulation of the material, the gel point was defined as the largest curing point, and the second-largest cluster achieved a peak [28,38]. The reaction process for the two systems is shown in Figure 4, with the black dashed line representing the gel point.

Despite these differences in DOP, the gel points of the two systems did not differ significantly, a phenomenon which shows that the two systems provide similar crosslinking processes, and the monomer size did not affect the crosslinking process directly.

### 3.2. Glass Transition Temperature (T_g_)

The glass transition temperature (*T*_g_) is one of the most critical properties of thermosetting polymers, as it directly influences their operational temperature range and constrains their potential applications. Polycyanurates, which typically exhibit *T*_g_ values in the 200–300 °C range, can be employed under demanding conditions. Three independent simulations were performed for each system following standard uncertainty quantification methods to investigate how monomer structure affects the *T*_g_ of these polymers. The results, along with their uncertainties, are summarized in Table 1. Figure 5 presents an example of the *T*_g_ determination for both systems. Two red lines were fitted to the density–temperature data in the high-temperature and low-temperature regions, as described in Section 2.5. The intersection of these two lines defines *T*_g_. We found that the S-large system exhibited a *T*_g_ of approximately 420 K, roughly 100 degrees lower than the S-small system’s *T*_g_ of about 520 K. Further, we analyzed the unreacted functional groups within the largest cluster, and the results revealed that the S-large system retained about 50% more unreacted groups than the S-small system (19.9 vs. 13.7%). The *T*_g_ results for all models in this study are available in the Appendix A.

### 3.3. Physical Properties

To understand the differences in mechanical responses between the two systems, we estimated their Young’s modulus under uniaxial tensile deformation at 1 atm and 300 K, following the procedure described in Section 2.5. The deformation tests were applied along the x-, y-, and z-axes, and the final Young’s modulus for each system was taken as the average over these three directions. A representative example of the stress–strain behavior for the two systems is shown in Figure 6. During testing, the maximum tensile strain was set to 15%. The Young’s modulus was obtained from the slope of the linear region of the stress–strain curve up to 5% strain. The results, summarized in Table 1, indicate that the S-small system exhibited a higher Young’s modulus of approximately 2.32 GPa, whereas the S-large system showed a notably lower value of about 1.77 GPa.

### 3.4. CO_2_ Adsorption and Free Volume

Polycyanurates naturally maintain microporous structures formed during their crosslinking processes, making them promising candidates for gas adsorption applications. As described in Section 2.5, we employed a combined GCMC/MD approach to predict CO_2_ adsorption in each system. The GCMC method enables the insertion or deletion of CO_2_ molecules in energetically favorable positions, while the subsequent MD simulations ensure structural relaxation and allow the system volume to adjust accordingly. We performed 40 GCMC/MD cycles for each model to achieve full uptake. The average results for all models are presented in Figure 7, where the shaded regions represent the standard errors. The S-small system is shown in blue, and the S-large system is shown in gray.

Both systems exhibited an initial increase in CO_2_ uptake over the first five cycles, followed by a plateau, indicating that the GCMC/MD method can accurately simulate a CO_2_ uptake behavior similar to that of real systems [7]. However, the S-small system demonstrated a significantly higher CO_2_ uptake (approximately 4 wt%) compared to the S-large system (about 1.3 wt%). Also, we calculated the free volume for the two systems; we found that the two systems only provided a 0.3% difference in free volume which cannot explain such a difference in CO_2_ adsorption values.

To further investigate the nature of this adsorption, we calculated the interaction energy between a CO_2_ molecule and the triazine ring at the B3LYP-D3/6-311+G(d) level of theory. Figure 8 highlights the interaction regions in red. The strongest interaction arose from the nitrogen atoms in the triazine ring, which interacted with the CO_2_ molecule’s carbon at approximately 2.94 Å. In contrast, the distance between the CO_2_ carbon and the benzene ring was around 3.5 Å. The adsorption energy, calculated as described in Equation (2), was approximately 6.7 kcal/mol. This relatively low adsorption energy explains why CO_2_ can be easily adsorbed and desorbed from polycyanurate materials [11]; the model information is available in the Appendix A.*E*_(interaction)_ = *E*_(CO2)_ + *E*_(triazine)_ − *E*_(com)_(2)

## 4. Discussion

For the crosslinking process and physical chemistry results for the two different systems listed in Table 1, we found no significant differences in density between the two systems and suggest a similar distribution of triazine rings. Both align well with experimental values from similar systems (1.195 ± 0.003 g cm^–3^) [39]. Also, this implies that the larger monomer does not provide a significantly more porous network or additional void spaces relative to the smaller monomer, thus confirming their structural similarities. However, the degree of polymerization (DOP) in the S-large system was approximately 10% lower than in the S-small system. This reduction can be attributed to the additional moieties near the cyanate functional groups in the larger monomer which hinder the approach of other cyanate groups and reduce the overall reaction probability. In the case of the gel point, the results are shown in Table 1. This finding indicates that the gel point has a weaker correlation with the thermosetting properties of systems composed of structurally similar monomers.

In the case of *T*_g_, given that both systems displayed similar DOP and the largest cluster sizes, we attribute the difference in *T*_g_ between the two systems (522 K vs. 420 K) primarily to the monomer size. The additional steric bulk near the cyanate groups impedes the approach of other reactive species, lowers the overall reaction probability, and promotes a more decoupled polymer chain architecture—ultimately resulting in a reduced *T*_g_. In addition, we analyzed the cluster geometric structure and compared the unreacted functional groups of the two systems. This increased fraction of unreacted sites (around 6.2% higher for S-small) was likely due to the enhanced steric hindrance that the larger monomer structure introduces, proving that the overall reaction probability is lower and making the polymer chains keep away from themselves.

We analyzed the Young’s modulus for the system’s mechanical properties, and the results show that S-large provided a lower material strength than S-small, with a 0.55 GPa difference. We attribute this difference primarily to the 7.1% higher degree of polymerization (DOP) in the S-small system, leading to longer polymer chains and thus enhancing mechanical strength with more flexibility [40]. In contrast, with its more branched structure, the S-large system tended to form shorter polymer chains and, consequently, displayed a reduced mechanical strength.

Regarding CO_2_ adsorption, by comparing the uptake process for the two systems, we found a significant difference from the initial stage, indicating that monomer size is a critical factor for gas adsorption. As shown in the shadow area, the standard error was smaller than the CO_2_ uptake values, proving that our models are repeatable and robust for gas adsorption prediction. For the final uptake values, given the comparable structures of the two systems and their modest 10% difference in DOP, we attribute this disparity primarily to the additional side groups present in the S-large monomers instead of the lower free volume in S-large because of the tiny difference in free volume, with only a 0.3% difference. These extra moieties reduced the free volume around the triazine rings, diminishing the available adsorption sites for CO_2_. By contrast, CO_2_ adsorption in polycyanurates was predominantly associated with the triazine ring environments rather than the benzene rings.

As per the QC calculation results of the interaction between CO_2_ and triazine ring, the distance between the carbon atom in CO_2_ and the nitrogen atom in the triazine ring was 2.94 Å; this proximity is attributed to dipole–quadrupole interactions. In contrast, the longer distance between the CO_2_ carbon and the benzene ring reflected weaker π–π interactions. The adsorption of CO_2_ molecules was not only affected by the dipole–quadrupole interactions, but also by a sub-interaction between benzene structures.

In the future, we will explore various blends of monomers using the approach described in this study. While it is impossible for simulations to perfectly replicate real systems and produce identical results to experiments, incorporating several critical factors into the simulations can still provide valuable insights. This strategy helps us better understand the structure–property relationships and, ultimately, supports the development of new materials.

## 5. Conclusions

This study employs two cyanate monomers with differing steric hindrance to prepare S-large and S-small systems. These systems are used to construct crosslinked polymer models using a previously reported algorithm, enabling MD simulations to predict their physicochemical properties and CO_2_ adsorption behaviors. While the gel points of both systems are similar, indicating that monomer size has a minimal impact on the crosslinking process, differences in their mechanical and thermal properties are evident. The S-large system exhibits a lower *T*_g_ and Young’s modulus than the S-small system. CO_2_ adsorption calculations, performed via GCMC/MD simulations, produce uptake curves that resemble realistic scenarios, highlighting the importance of certain structural features in governing gas adsorption. Subsequent QC calculations validate the identified CO_2_ adsorption sites, demonstrating that CO_2_ molecules can be readily adsorbed and desorbed from the polymer network.

In this study, we report on the effect of monomer size on the final polymer’s physical properties, and a bigger monomer provides a lower *T*_g_ and Young’s modulus by decreasing the reaction probability of one side of the cyanate group. Also, this research explains that some critical positions in the polymer greatly affect CO_2_ adsorption. By extending the free volume around these positions or increasing the density of critical positions [41], the CO_2_ adsorption values can be controlled. The results of this study confirm that the modeling approach reported by our previous research can predict the physical properties of polymers with different monomers and provide a possible approach to controlling CO_2_ adsorption in polymer materials. In our future research, we will extend this approach to explore a broader range of monomers with different structural and chemical characteristics, aiming to enhance CO_2_ adsorption capabilities and inform the development of next-generation polymeric materials for carbon capture applications. This study enhances our understanding of the structure–property relationships in polymer materials, providing valuable insights into industrial polymer production.

## Figures and Tables

**Figure 1 polymers-17-00148-f001:**
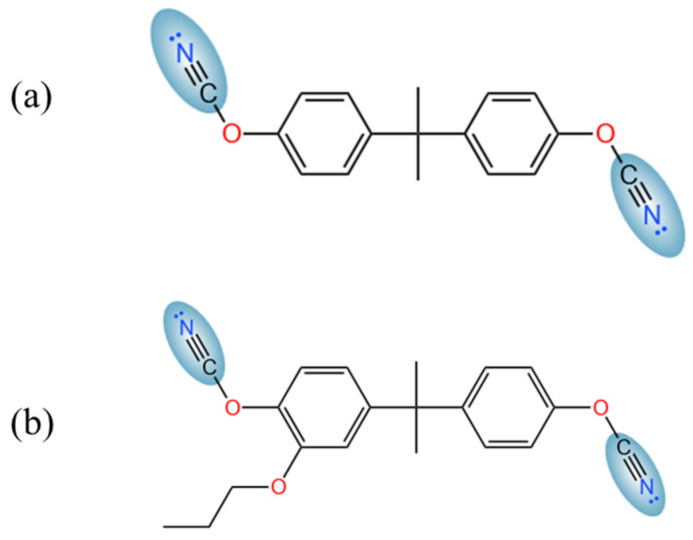
Two cyanate monomer molecular formulas used in this study. (**a**) commercial cyanate monomer for S-small; (**b**) cyanate monomer with extra functional group for S-large.

**Figure 3 polymers-17-00148-f003:**
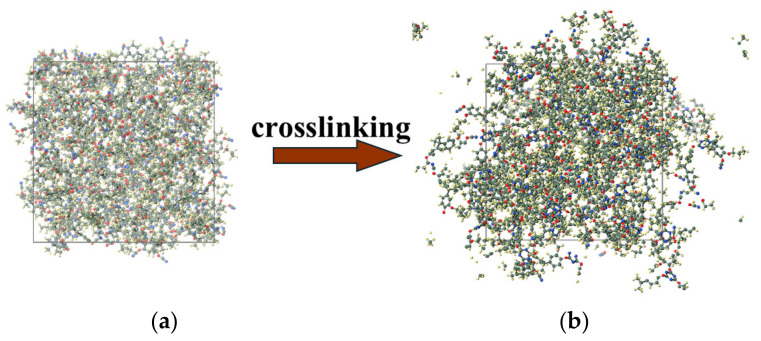
Crosslinking process from the (**a**) initial state to the (**b**) final state. In the initial state, only the catalyst molecules were opaque. In the final state, all crosslinked molecules were opaque (purple: zinc; grey: carbon; red: oxygen; white: hydrogen; blue: nitrogen).

**Figure 4 polymers-17-00148-f004:**
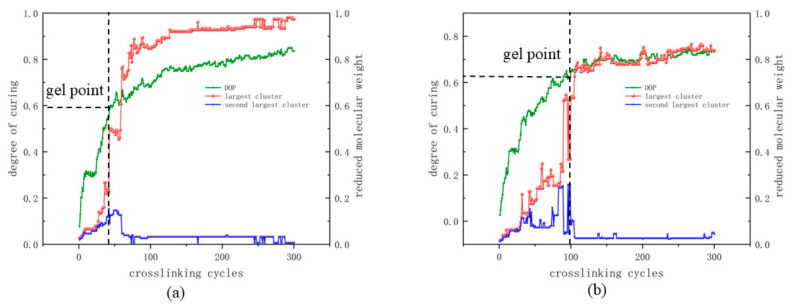
The crosslinking process for two different systems: (**a**) S-small and (**b**) S-large. The red line is the largest cluster; the blue one is the second-largest cluster; the green line is for DOP. The black dashed line is used to point out the gel point.

**Figure 5 polymers-17-00148-f005:**
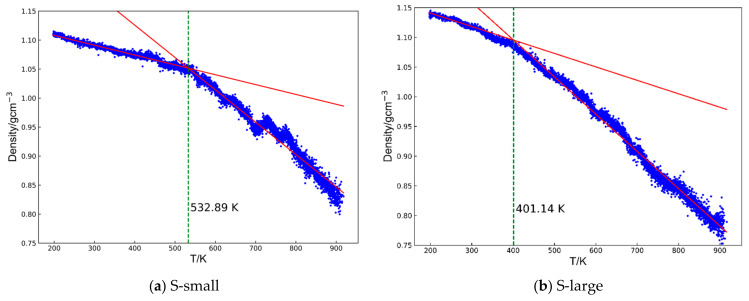
The representative output of the volume–temperature plot for (**a**) S-small and (**b**) S-large consists of the curve being fitted by two straight lines representing the two states, glassy and elastic. The cross-point temperature for two lines is determined as the glass transition temperature, as indicated by the green dashed line.

**Figure 6 polymers-17-00148-f006:**
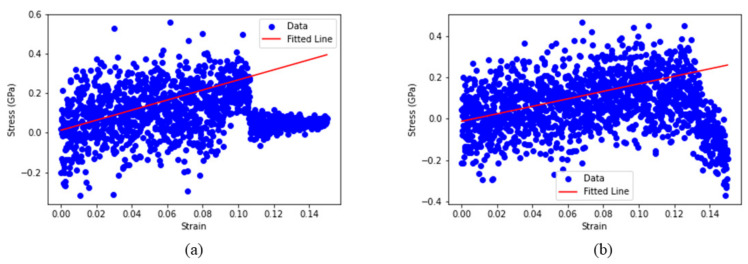
The representative output of the stress–strain plot for (**a**) S-small and (**b**) S-large, with the curve fitted by a straight red line based on the least squares method.

**Figure 7 polymers-17-00148-f007:**
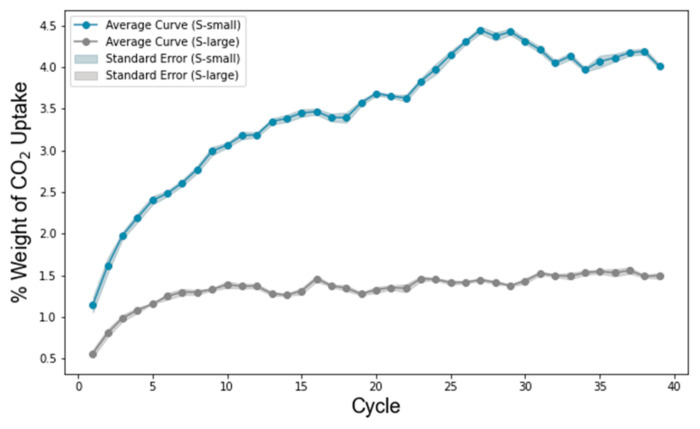
Total average weight ratios of the CO_2_ uptake of different systems predicted via the GCMC/MD method. Color code: blue, S-small; gray, S-large.

**Figure 8 polymers-17-00148-f008:**
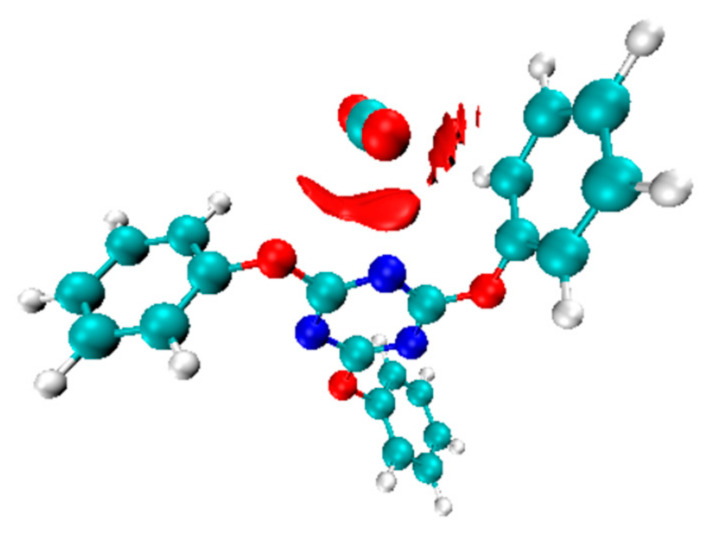
The interaction between CO_2_ molecules and the triazine ring structure. Red clouds represent the interaction area between CO_2_ molecules and the triazine ring: with more clouds, the interaction is stronger. Color code: green, carbon atom; white, hydrogen atom; blue, nitrogen atom; red, oxygen atom.

**Table 1 polymers-17-00148-t001:** Thermophysical properties of BPACN systems are crosslinked via the two different monomers.

	S-Small	S-Large
Density [g cm^–3^]	1.102 ± 0.004	1.112 ± 0.004
DOP [%]	83.8 ± 0.3	76.7 ± 1.4
Unreacted functional groups in the largest cluster [%]	13.7 ± 1.1	19.9 ± 0.76
Gel point [%]	65.2 ± 1.7	62.4 ± 2.6
*T_g_* [K]	522.0 ± 8.4	420.7 ± 18.5
Young’s modulus [GPa]	2.32 ± 0.23	1.77 ± 0.18
Free volume [%]	0.88 ± 0.01	0.52 ± 0.03

## Data Availability

The original contributions presented in this study are included in the article. Further inquiries can be directed to the corresponding author.

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
