# Peer review of "Influence of Monomer Size on CO2 Adsorption and Mechanical Properties in Microporous Cyanate Ester Resins"

_polymers, 2025, doi:10.3390/polym17020148_

Round 1

Reviewer 1 Report

Comments and Suggestions for Authors

This paper seems good with potentiality. However, the present state of the paper can be improved by addressing the following comments-

1.       Background discussion in abstract should be minimized.

2.       Authors should improve abstract with more quantitative data.

3.       The paper lacks novelty. Authors should mention the novelty of this paper.

4.       Discussion part should be improved.

5.       Impacts of results should be discussed more elaborately.

6.       What are the significances of the findings in industry?

7.       Most of the papers authors have cited are old. They should cite more recent papers. 

Reviewer 2 Report

Comments and Suggestions for Authors

Comments

The manuscript reported the design and analysis of cyanate resin models with varying monomer sizes, focusing on their crosslinking processes, physicochemical properties, and COâ‚‚ adsorption behaviors. Key findings include the negligible impact of monomer size on crosslinking but notable effects on properties like glass transition temperature and Young’s modulus. COâ‚‚ adsorption was influenced by specific adsorption sites, highlighting a strategy for optimizing thermosetting polymers for gas separation and storage applications. After a thorough review of your manuscript, I have identified several areas that require significant revision to enhance the clarity, depth, and impact of your work. Please consider the following specific points:

1.     The manuscript asserts that monomer size does not significantly affect the crosslinking process but does influence properties like glass transition temperature and Young’s modulus. However, the underlying mechanisms for these observations are not sufficiently explained. Please provide a detailed analysis or theoretical justification elucidating how monomer size impacts these specific physicochemical properties.

2.     Quantum Chemistry Calculations and GCMC/MD Approach: The manuscript references the use of quantum chemistry calculations and a GCMC/MD approach to investigate COâ‚‚ adsorption. Elaborate on the computational parameters, models, and software utilized in these calculations. Additionally, discuss the convergence criteria and any validation steps taken to ensure the reliability of your simulation results.

3.     Although the author mentions that the introduction of propyl ether groups reduces COâ‚‚ adsorption, how can further optimization be achieved? Could chemical modification or the design of new functional groups enhance adsorption performance, thereby providing more guidance for practical applications?

4.     English writing needs to be re-corrected carefully, including some grammar and spelling errors.

5.     The article contains numerous formatting errors, such as multiple instances where COâ‚‚ is not subscripted and issues with the formatting of Tg. Additionally, the references section has numerous formatting errors that need to be corrected.

6.     The description of Figure 1 is missing some content. In Figure 3, the labels “a” and “b” are missing.

7.     The clarity of the figures throughout the article needs to be improved.

Overall, the organization and writing of the manuscript are too preliminary to consider for publication. I suggest resubmitting it after thorough revisions.

Reviewer 3 Report

Comments and Suggestions for Authors

pls see pdf attached

Round 2

Reviewer 1 Report

Comments and Suggestions for Authors

Necessary corrections have been made. This paper can be published in its present form. 

Reviewer 2 Report

Comments and Suggestions for Authors

The revised version has addressed all of my suggestions, which should be accepted for publication.

Reviewer 3 Report

Comments and Suggestions for Authors

I am satisfied with the revisions